# Effect of Cannabis on Memory Consolidation, Learning and Retrieval and Its Current Legal Status in India: A Review

**DOI:** 10.3390/biom13010162

**Published:** 2023-01-12

**Authors:** Nandi Niloy, Tousif Ahmed Hediyal, Chandrasekaran Vichitra, Sharma Sonali, Saravana Babu Chidambaram, Vasavi Rakesh Gorantla, Arehally M. Mahalakshmi

**Affiliations:** 1Department of Pharmacology, JSS College of Pharmacy, JSS Academy of Higher Education & Research, SS Nagar, Mysore 570015, Karnataka, India; 2Centre for Experimental Pharmacology and Toxicology, Central Animal Facility, JSS Academy of Higher Education & Research, SS Nagar, Mysore 570015, Karnataka, India; 3Department of Anatomical Science, St. George’s University, University Centre, St. Georges FZ818, Grenada

**Keywords:** tetrahydrocannabinol, cannabidiol, consolidation, learning, retrieval, legal India

## Abstract

Cannabis is one of the oldest crops grown, traditionally held religious attachments in various cultures for its medicinal use much before its introduction to Western medicine. Multiple preclinical and clinical investigations have explored the beneficial effects of cannabis in various neurocognitive and neurodegenerative diseases affecting the cognitive domains. Tetrahydrocannabinol (THC), the major psychoactive component, is responsible for cognition-related deficits, while cannabidiol (CBD), a non-psychoactive phytocannabinoid, has been shown to elicit neuroprotective activity. In the present integrative review, the authors focus on the effects of cannabis on the different cognitive domains, including learning, consolidation, and retrieval. The present study is the first attempt in which significant focus has been imparted on all three aspects of cognition, thus linking to its usage. Furthermore, the investigators have also depicted the current legal position of cannabis in India and the requirement for reforms.

## 1. Introduction

*Cannabis sativa*, commonly known as cannabis (Family: Cannabaceae), is one of the oldest crops cultivated and has various religious and medicinal values [1]. Nondrug assortment of cannabis is popularly known as hemp, often used in food and textile industries [2].

The genus Cannabis is taxonomically classified into three different species: *Cannabis sativa*, *Cannabis indica*, and *Cannabis ruderalis*. The sativa and indica species are primarily developed owing to their medicinal and commercial benefits compared to ruderalis [3]. Cannabis is known to yield more than 400 chemicals [4] of different classes, out of which 144 are cannabinoids with pharmacologically varying effects [5]. The most notable ones are tetrahydrocannabinol (THC) and cannabidiol (CBD). THC is the major psychoactive compound, while CBD is the nonpsychoactive moiety [6].

The medicinal use of cannabis was depicted in India and China much earlier than its introduction in Western medical systems for treating various ailments such as fever, urinary disorders, inflammation, meningitis, gastrointestinal, and dermatological problems [7,8,9]. Later, during the 19th century, multiple extracts and tinctures of cannabis were used to treat pain, whooping cough, and asthma, and used as a sedative or hypnotic [10,11].

Modern studies have shown that cannabis has numerous beneficial effects in neurodegenerative disorders of Alzheimer’s, Parkinson’s, epilepsy, Huntington’s, multiple sclerosis, and amyotrophic lateral sclerosis [11,12,13,14,15]. Reports have also shown that cannabis possesses anti-inflammatory, anti-depressant, anti-anxiety, anti-psychotic, and anti-schizophrenic effects [16,17,18]. Similar neuroprotective effects have also been exhibited by synthetic cannabinoids HU-211 and endocannabinoids anandamide (AEA) [19,20]. Despite the numerous medicinal properties of cannabis, stringent government policies regulating the use of cannabis have almost made it impregnable to research activities. Neurodegenerative disorders currently concern the world, affecting memory processing (learning, consolidation, and retrieval process). Moreover, studies and reviews have focused on memory, but not many studies have discussed the entire memory process. Therefore, in this current integrated review, an attempt is made to summarize the reports demonstrating the effects of cannabis on memory function, focusing on the learning, consolidation, and retrieval processes. The paper also encompasses details on the current legal position regulating the access, handling, and use of the herb in India.

## 2. History of Cannabis: Background and Religious Attachments

Archaeobotanical evidence suggests the ritualistic and medicinal use of cannabis since 800 BCE. Assyrians, Jews, Arabians, and the Greeks were the first to adopt cannabis use for medicinal purposes, as evident from the ancient texts of the respective communities [21]. Early evidence displayed cannabis use in Central and Southeast Asia dating back to 11,700 years ago. The relocation of the Scythians from Central Asia led to the influx of cannabis to Europe and the Mediterranean regions [21]. In China, the evidence of cannabis use for food and clothing purposes was found in a later Neolithic site, dating back to 5000 BC [22]. Fossilized evidence of seeds and fruits suggest the existence of cannabis in Japan dating back to 10,000 years ago, during the Jomon Period [23]. Hemp remains (1303–1212 BCE) suggest Egyptians as the early users of cannabis in the African subcontinent. The movement of African slaves during the 16th and 17th century led to the disposition of the herb in South America. Evidence of wild-type cannabis was found in European countries, such as Hungary and Bulgaria, dating around 10,200–8500 years BC [21,24].

In India, the sacred scripture “Atharva Veda” claimed cannabis as an herb of happiness, as it elicited joy and pleasure, and hence used in ritualistic activities. The Ayurvedic system of medicine also described the use of cannabis for treating various gastrointestinal, respiratory, and urinary disorders [6,24,25]. The religious amalgam of cannabis was noticed in Tibetan practices and Buddhism together with Hinduism [26]. Tantric traditions also showed the unorthodox use of cannabis for different ceremonial proceedings, upholding the history of cannabis in India [27].

## 3. Modern Medical Science

### Endogenous Cannabinoid System

Endogenous ligands that interact with cannabinoid receptors, affecting behavior similar to the effects produced by the psychoactive component (THC) of cannabis, are known as endogenous cannabinoids. In addition to the endogenous substances like virodhamine and 2-arachidonoyl glycerol ether [28], the most well-characterized and studied ligands are anandamide (AEA) and 2-arachidonoyl glycerol (2-AG). AEA has high affinity and imparts partial agonistic activity at CB1 and CB2 receptors. On the other hand, 2AG has moderate to low affinity at both cannabinoid receptors with complete agonistic activity. Both endocannabinoids are formed, transported, and degraded by different mechanisms [29,30].

AEA is synthesized by the catalytic action of N-acyl-phosphatidylethanolamine-specific phospholipase D (NAPE-PLD) on N-acyl-phosphatidylethanolamine (NAPE), whereas 2AG is synthesized by the action of diacylglycerol lipase (DAGL) α or β on DAG. Following the release into intracellular space, they travel via proposed mechanisms of simple diffusion (concentration gradient), endocytosis (lipid rafts), or via carrier proteins (fatty acid binding proteins and heat-shock protein 70). After the cellular uptake, AEA and 2AG are degraded by fatty acid amide hydrolase (FAAH) and monoacylglycerol lipase (MAGL), respectively [29,30,31], as shown in Figure 1. The formation of NAPE and DAG is considered to be the rate-limiting and Ca^2+^-sensitive step in the entire process. Both endocannabinoids have different receptor selectivity but are, however, produced in response to the increased intracellular calcium concentrations [32].

Increased calcium concentration or activation of Gq/11-coupled receptors leads to the majority involvement of 2 AG in the retrograde signaling process following the release of 2-AG into the extracellular space [33,34], leading to its (2-AG) arrival at the presynaptic terminal where the interaction with CB1 receptor triggers the inhibition of neurotransmitter release. However, cessation of signaling occurs by the degradation of 2-AG by MAGL, which is expressed in selective synaptic terminals and glial cells [35].

## 4. Cannabinoid Receptors

Cannabinoid receptors are widely distributed in all the major organ systems such as the GI tract, liver, muscle, pancreas, adipose tissues, brain, and lungs. There are two major cannabinoid receptors—cannabinoid 1 (CB1) and cannabinoid 2 (CB2) [35]. Other than CB1 and CB2, endocannabinoids also show interaction with G-protein coupled receptors (GPR3, GPR6, GPR12, GPR18, and GPR55), transient receptor potential vanilloid-1 (TRPV1) channels, and peroxisome proliferator-activated receptors (PPARs) [36,37]. A study by Sait et al. (2020) through high-resolution X-ray crystallography demonstrated that cannabidiol interacts with NavMs voltage-gated sodium channel by binding to a site which is an interface of fenestrations and central hydrophobic cavity [38]. A fluorescence-based thallium flux assay showed that cannabidiol enhances the flux through human KV7.2/7.3 channels at submicromolar concentrations, thus shifting the channels to hyperpolarized state by enhancing the native M-current in mouse cervical ganglion and rat hippocampal region [39]. Another study demonstrated the ability of cannabidiol to counteract the pathogenic effects of Kv7.2 variant channels, thus establishing the use of cannabidiol as a potential therapeutic agent to reduce seizure activity [40].

### 4.1. Cannabinoid 1 (CB1) Receptors

In the central nervous system, high CB1 distribution is present in regions of olfactory bulbs (inner granule and inner plexiform layer) [41,42], forebrain regions of cerebral neocortex frontal lobe (II, III, V, VI layers of frontal lobe) [41], regions of the hippocampus (stratum pyramidale), dentate Gyrus (granule cell layer), and amygdala [41,43,44]. The receptor distribution varies from low to moderate in the thalamus (low), hypothalamus, and nucleus accumbens [45,46,47].

The basal forebrain holds moderate CB1 receptor distribution where basal ganglia (globus pallidus) consist of the highest CB1 receptor expression for the same region [41,44].

In the midbrain, immunostaining showed high CB1 expression in substantia nigra pars reticulata, oculomotor nucleus, and red nucleus, whereas periaqueductal gray (PAG) had moderate levels of the receptor expression concentrated on the GABAergic neuron terminals [41,46].

Concerning the hindbrain, CB1 distribution was found to be high within the cerebellar cortex (molecular layer), and strong immunostaining was reported around the Purkinje cell bodies (basal portion) with low positive staining in the granule cell layer [41,45]. In addition, high immunostaining was also evident within the medulla oblongata (cochlear and trigeminus nucleus) [41].

In the spinal cord, low levels of CB1 receptors were expressed in the brainstem region [44]. Staining revealed that spinal cord areas of the dorsal horn, intermediate region, and ventral horn show high CB1 immunoreactivity, whereas dorsal root ganglia (DRG) expresses low to moderate CB1 expression [41,48].

Considering the peripheral nervous system, CB1 receptors are strongly expressed within the thoracic spinal nerve, depicting high expression in Schwann cells [41].

The enteric nervous system (enterochromaffin cells, lamina propria, epithelial cells, goblet cells, gastric mucosa, myenteric plexus, submucosal plexus neurons) [49,50,51] and reproductive system [52,53] also have their share of CB1 receptors and are modulated by CB1 activation.

Recent studies also revealed that the expression of CB1 receptors is not restricted to the plasma membrane but also to the outer mitochondrial membrane in the brain. Localization of CB1 receptors in both pre- and postsynaptic axonal terminals accounts for about 15% of the total brain CB1 receptors [54]. Expression of CB1 receptors in astrocytes also explains the role of CB1 receptors in neuroinflammation and neurotransmission [55].

### 4.2. Cannabinoid 2 (CB2) Receptors

Previous studies confirmed the presence of CB2 receptors in the spleen and immune cells, and were believed to be restricted to these sites [56,57]. Since they are expressed in the immune cells, their presence is also anticipated in the brain microglia. Dense CB2-positive immunoreactivity was also found in the retina, areas of the anterior olfactory nucleus, cerebral cortex, hippocampus, amygdala, striatum, thalamic nucleus, PAG, substantia nigra, cerebellum, pons, and medulla [58,59].

Recent immunohistochemical assessment of rat brain also revealed the presence of CB2 receptors in the brain glial cells [60] and neuronal cells, including the ventral tegmental area (VTA), dopamine (DA) neuron, prefrontal cortical neurons, and hippocampal CA3/CA2 pyramidal neurons [31,61,62].

Unlike CB1 receptors, the postsynaptic presence of CB2 receptors was also found in the cerebellar Purkinje (intense) and dendritic cells (moderate); thus, its activation leads to the inhibition of postsynaptic neuronal activity. However, the presynaptic distribution of the CB2 receptors was predicted due to the presence of the receptor in the unmyelinated axons of substantia nigra pars reticulata [63].

CB2 immunostaining was also observed in the digestive mucosa, muscular layers, intestinal submucosal plexus, and glial cells [49]. Research reports also showed the involvement of CB2 receptor expression in conditions of glioblastoma, brain tumors, schizophrenia, Parkinson’s, and Alzheimer’s. The abovementioned evidence depicts that CB2 are not only restricted to peripheral sites but also to the brain; however, the expression level is much lower than that of CB1 receptors, which suggest that CB2 receptors might not indulge in centrally mediated functions, unlike CB1, which mediate the psychoactive effects of cannabinoids upon activation of sodium channels [64,65,66,67,68]. Recent studies also revealed the expression of CB2 receptor mRNA in T and B lymphocytes, eosinophils, neutrophils, and monocytes, eliciting the need to uncover the role of CB2 receptors as immunoregulators [69].

## 5. Cannabis and Cognitive Functions

The role of cannabis on cognitive functions is a matter of long debate. Endogenous cannabinoids (AEA and 2 AG) are involved in memory and learning (including rewarding processing) through pre- and postsynaptic modulation (crucial for learning) [70,71,72]. CB1 receptors are involved in the modulation of the excitatory and inhibitory neurotransmitter (glutamate, GABA, and DA) activities [73]. The basal ganglionic and cerebellar presence of endocannabinoids and its mediated interaction of CB1 receptors with DA elicits their additional role in cognition and emotional regulation [74]. A recent study displayed the social, behavioral, and communicational alterations similar to the features established in autism spectrum disorder (ASD) in the CB1 knockout mice, indicating the critical role of CB1 receptors in core behavioral domains [75]. In contrast, another study demonstrated that genetic disruption of CB1 receptor gene resulted in improvement of behavioral habituation [76].

Cannabis intake during the early days in one’s life was correlated with minimized IQ [77], mounting likelihood of schizophrenia [78], and disturbed active memory [79]. Previous literature demonstrated the detrimental role of cannabis in various neurodegenerative disorders (Alzheimer’s, Parkinson’s, Huntington’s, dementia, and epilepsy [12,80,81,82]. Studies have also suggested the onset age for cannabis use is very important in determining the impact on the cognitive dimension [83,84]. Participants who began the use before 18 years were more cognitively impaired than those who entertained the use in their later years [79]. Moreover, cannabis produced higher detrimental effects on cognition in teens than alcohol, affecting the neuronal brain tissue accountable for memory [85]. A study by Cohen et al. (2021) showed that THC exposure in adolescent rats affected memory and plasticity through the hippocampal–accumbens pathway [86]. Another study, by Tagne et al. (2021), demonstrated that frequent exposure to THC in male mice during adolescence resulted in dormant dysfunction in social behavior [87]. Additionally, chronic exposure to THC in adult rodents resulted in depressive behavior and suicidal tendency, which is correlated to the reduced serotoninergic neural activity in the young rats compared to the adult rats [88]. These data indicated that adolescents were more highly vulnerable to THC exposure than adults.

Further, a randomized controlled trial on the acute and delayed effects of THC intoxication on the susceptibility to false memory in 64 healthy volunteers evidenced enhanced false memory in the THC-intoxicated subjects. It was also reported that false memory effects were mostly observed in the acute-intoxication phase [89]. Additionally, another field study conducted by Kloft et al. (2019) revealed that cannabis-intoxicated and sober cannabis consumers showed false memory compared to control participants [90].

Memory dissociative effects of cannabis were correlated to the duration and frequency of consumption [79,91,92]. Extended memory disassociation was further correlated with extended exposure to cannabis [93,94,95,96]. Studies showed that cannabis-mediated effects persisted even after 28 days of withdrawal [84,93]. In summary, the available evidence supports the involvement of endocannabinoids in learning and cognitive processes, providing an insight that exogenous administration of cannabis might negatively modulate cognitive activities; however, contradictions exist while considering the use of cannabis as a medicinal product.

### 5.1. Cannabis and Learning

In the context of learning, a relationship was established between learning and long-term potentiation (LTP) and long-term depression (LTD), and further, the relationship was shown to be adversely impacted by THC [97,98,99,100,101]. Preclinical data revealed that several THC-mediated learning deficits majorly mediate via a CB1-dependent mechanism in various cases of learning assessments and related tasks [102,103,104,105,106,107]. Blockade of the CB1 receptors increased gene expression of BDNF, Gria1, and Syn1 linked to increased synaptic plasticity [108]. Furthermore, blockade also improved acquisition learning and reversal learning, leading to overall learning improvement [109]. MRI imaging of PFC in heavy cannabis users revealed decreased size and low PFC activity attributed to the disruption of endocannabinoid-mediated synaptic plasticity [110,111,112], suggesting the cannabis interference with the physiology of the PFC and, thus, the learning process. Similarly, fMRI reports displayed low verbal learning and poor error learning capability associated with cannabis-related hypoactivation of the hippocampus, midbrain, and dorsal anterior cingulate cortex, highlighting the consequence of drug abuse in learning-related aspects [113,114]. Interaction of cannabinoid agonists with neurotransmitters (dopamine, GABA, and acetylcholine) in the hippocampal CA1 region elicits stimulation of state-dependent learning (same state of consciousness as it was when memory was formed) [115,116,117].

Marijuana exposure during the adolescent stage impacted the reactivity via THC with CB1 receptors interaction in the major brain areas of the hypothalamus, basal ganglia, cerebellum, prefrontal cortex (PFC), and cingulate gyrus [118,119,120]. Therefore, the cognitive activities served by these regions might become mostly affected by early cannabis exposure and account for the problems of learning and memory. Exposure to marijuana in this period hampered academic-associated learning [111,121,122], in addition to social and professional hurdles [123]. Cannabis exposure also induced neurological changes in the brain structure, reducing the gray [124] and white matter [125,126,127] integrity, accompanied by degraded neurocognitive performance [128,129]. However, withdrawing the use for 72 h or a month improved memory, reducing use-related deficits, thereby suggesting temporary effects of cannabis use [130,131]. Findings from Human Connectome Project revealed the detrimental effects of cannabis exposure on learning analysis in a larger population [132]. Previous research tried to identify the neurocognitive effects of chronic cannabis exposure associated with MDMA (3, 4-methylenedioxymethamphet-amine), which was linked to inadequate performance in learning and memory tests, proving an important covariant in MDMA-related cognitive deficits [133]. Similarly, a meta-analysis also reported learning deficit to be associated with chronic cannabis exposure [134].

Outcomes also revealed the detrimental effects of cannabis (THC-rich content) exposure on the learning process. A few cases displayed gender-biased effects, where female rats were found to be more sensitive to THC [105,135]. Clinical data also depicted the same detrimental effects of THC on learning [136,137,138]. On the other hand, cannabidiol (CBD) mitigated THC-induced learning impairment [139] along with eliciting memory-rescuing effects in various neurodegenerative diseases [140,141]. CBD also suggested improvement in learning and memory by increasing the dendritic spine densities involved in synaptic plasticity [142,143]. Additionally, CBD improved spatial learning [144] and verbal learning along with enhanced learning following brain damage [145,146]. Therefore, CBD holds great potential for treating learning- and memory-related deficits. Further, it is also an approved drug by USFDA in treating conditions such as idiopathic epilepsy, CDKL5 deficiency, resistant Lennox–Gastaut syndrome, Dravet syndrome, and tuberous sclerosis complex.

Nevertheless, contradictions exist as the connection between cannabis use and cognitive alterations is complex [147]. Reports showed low THC dose improved learning and cognition, suggesting dose-dependent effects of THC in vivo [148,149,150]. Moreover, exposure to aquatic cannabis extract (THC) in rats significantly decreased learning time [151,152]. In addition, some reports were inconclusive and insignificant as no differences were observed in the gray and white matter on cannabis exposure [147,153,154,155]. However, the debate still continues as cannabis treatment improved cognition in several aspects [145,149]. The dose range certainty is required to establish to observe any significant effects in clinical trials.

### 5.2. Cannabis and Memory Consolidation

Memory consolidation is a time-dependent mechanism by which a temporary, vulnerable memory is evolved to a more stable and long-lasting form and becomes immune to intrusion from competing influences in the absence of further practice [156]. Memory continues to mature between encoding and recall via the consolidation process [157].

The overall endocannabinoid system plays a modulatory role in the coherence of neuronal activity in the brain regions (hippocampus, amygdala, PFC), leading to consolidation of aversive or arousing experiences through CB1 activation [158,159,160], thereby suggestive of the involvement of endocannabinoids in memory consolidation.

Emotional stress and aversive and arousal events stimulated neurological systems that played an integral role in memory consolidation and maintaining the strength of memories in accordance with their emotional significance [161]. Garcia et al. (2016) explored the role of CB1 receptors on stress-induced impairment of nonemotional memory consolidation. Interestingly, the study revealed that both central and peripheral mechanisms are involved in the stress-induced object-recognition memory impairment acting through CB1 receptors of adrenergic and nonadrenergic cells. The study by Busquets-Garcia et al. also opened newer avenues in the treatment of stress-related cognitive aspects [162].

Cannabinoids and CB1 receptor agonists were also shown to modulate memory function by affecting memory consolidation [163,164,165]. A preclinical investigation by Morena et al. (2014) showed that infusion of URB597, a fatty acid amide hydrolase inhibitor, selectively increased AEA levels in the basolateral complex of the amygdala, hippocampus, and mPFC, and thus enhanced memory, suggesting the critical role of endogenously released AEA in memory consolidation [166]. Nevertheless, memory consolidation is prevented by CB1 antagonism mediated by intrahippocampal infusion of the receptor antagonist, thus hampering long-term memory consolidation by inhibiting LTP [167,168].

The prevention of 2AG degradation via the MAGL inhibitor enhanced the 2AG-mediated memory consolidation of aversive events via CB2 receptor activation [169]. Intra-amygdala CB1 agonist infusion disrupted fear memory by hampering the reconsolidation (past consolidated memories being recalled and actively consolidated) [170]. Administration of WIN55,212-2 (WIN, cannabinoid agonist) and JZL-184 (monoacylglycerol lipase inhibitor, increases endocannabinoid levels) within the cerebellar cortex supported memory consolidation [171]. Steinmetz et al. (2018) also demonstrated the beneficial effects of cannabinoid receptor agonists on motor learning by acting on the CB1 receptors in the cerebellar cortex [172].

In addition, WIN also attenuated fear consolidation in the basolateral amygdala (BLA) and not in the PFC, suggesting the differential regulation of fear memory consolidation [173].

Cannabidiol (CBD) also plays an integral role in memory consolidation. Direct CBD infusion into the PFC postfear conditioning (pairing aversive stimulus with a neutral context or stimulus) distorted the associative fear memory in rats [174]. Similarly, microinjections of CBD into the PFC postfear conditioning disturbed contextual fear memory consolidation, supporting the interfering role of CBD via reduced PFC dependency on the cortico-limbic region [175]. CBD also interfered with the reconsolidation of different fear memories (recent and older) via dorsal hippocampus CB1 and CB2 activation, as well as disturbed the contextual drug abuse-mediated memory reconsolidation (highlighting its therapeutic usage), and promoted lower relapse [176,177,178].

Similarly, intradorsal hippocampal CBD infusion distorted memory consolidation when delivering postfear modification, allowing extended time to reduce aversive memories after acquisition via involvement of anandamide, CB1, CB2, and PPARγ receptors in the process [179]. The above evidence suggested the involvement of cannabinoid receptor subtypes (PPARα, PPAR-γ, and TRPV1) in cannabinoid-intervened memory consolidation [180]. CBD alone significantly reduced fear memory reconsolidation, and a combination dose of CBD/THC/remaining plant extracts (RPE) of cannabis diminished the reconsolidation of learned fear. The effects of THC on the reconsolidation of fear were modulated by CBD and RPE [180]; however, these findings were contrary to earlier research, where THC alone disturbed the reconsolidation of fear memory [181]. Additionally, it is still unknown whether CBD can disrupt the reconsolidation of aversive memories and support their extinction (gradual decrease in response to conditioned stimuli) in humans [182]. CBD, when administered postextinction in participants, increased the consolidation of extinction memory (gradual decrease in conditioned response), suggesting CBD might be incorporated as an adjuvant to extinction-formulated remedies for anxiety ailments [183].

On the other hand, THC administration enhanced extinction learning (gradual decrease in reaction to a conditioned stimulus when delivered without consolidation) and consolidation via amygdala activation (involved in short-term extinction retention). The long-term effects of THC were obtained via increased activation of vtmPFC, suggesting the role of THC in accelerating the consolidation process [184]. However, a contradiction exists where no THC-mediated effects were observed on extinction learning but increased vtmPFC activation [185]. The difference in doses, regimen, THC tolerance levels, and route of administration might point out the reasons for the contradiction. However, studies concentrating on reconsolidating aversive or bad memories require more investigation. Therefore, the combination of CBD and THC or CBD alone could be advantageous therapeutically, providing a novel outlook for the neural mechanistic approach toward memory consolidation.

Memory retrieval is pivotal in determining whether humans can recall information stored in long-term memory. If more associations and links were focused on particular information, the information was effectively consolidated and recalled with ease [186]. Few studies demonstrated the negative effects of intracerebral and peritoneal administration of cannabinoid agonists on memory retrieval [163,187]. Hippocampal administration of WIN55, 212 (WIN, cannabinoid agonist) reduced memory retrieval in rats on the test day (step-down avoidance task), suggesting the cause to be a blockade of glutamate, acetylcholine, and noradrenaline release in the hippocampus via a CB1-dependent mechanism [188,189]. Similarly, microinjection of WIN and VDM-1 (endocannabinoid membrane transporter inhibitor) reduced memory retrieval immediately post-training [190]. Microinjection of WIN in the basolateral amygdala (BLA) blocked acquisition and consolidation but not retrieval, whereas in the ventral subiculum (VSub) interfered with the retrieval [191]. Moreover, WIN diminished retrieval in the BLA and PFC, suggesting that memory retrieval was modulated variably by the amygdala and cortical CB1 receptors [173].

Similar to the exogenous cannabinoids (WIN), endogenous cannabinoids, particularly hippocampal 2-arachidonoyl-glycerol (2- AG) (increased levels), are involved in memory processes, mainly regulating the spatial memory retrieval of stressful encounters [192]. The endocannabinoid system also plays a midway role in corticosterone-mediated retrieval impairment, where the blockade of the hippocampal CB1 receptor prevents the retrieval deficit. In contrast, the same impairing dose of corticosterone enhances the endocannabinoid 2 AG levels in the hippocampus, suggesting the modulatory effect of the endocannabinoid system in retrieval impairment [193]. A study by Morena et al. (2014) demonstrated that anandamide, an endogenous cannabinoid, was released during the aversive training into the amygdala, hippocampus, and medial prefrontal cortex, indicating the critical role of the prefrontal-limbic circuit in memory consolidation during emotionally aroused training [194].

Preclinical evidence shows that inhalation of cannabis smoke or THC interfered with the retrieval memory task in the Morris water maze (MWZ) [194,195]. Moreover, THC also increased false retrieval on both emotional and false memory tasks, thus producing unwanted effects during retrieval [196]. However, clinical trials for the same are in progress [197]. The effects of THC described above were in coherence with the findings of the study in which acute administration of THC dose dependently reduced memory retrieval, response rate, and increased percentage errors in both ovariectomized and intact rats [198]. Gender-specific differences were also observed in retrieval, where male offspring showed a greater decrease in novel object preference than females, affecting the recognition memory, which is crucial for information retrieval. However, within a similar concept, a contradicting 5-year survey showed females to be more sensitive to cannabis exposure and producing a higher degree of neurocognitive implementation, including delayed retrieval [199,200]. Thus, it could be understood that cannabis exposure produces gender specific effects.

Clinical THC administration influenced the successful retrieval of extinction memory (repeated exposure to a conditioned stimulus) more than the control [201]. In a clinical setup of virtual MWM, a retrieval deficit was observed among the cannabis users vs. nonusers, suggesting hypoactivation in the right parahippocampal gyrus and cingulate gyrus in users [202]. Concurrent use of cannabis and tobacco had effects on the neurocognitive profile, linking cannabis use with poor retrieval and acquisition in a verbal learning analysis prevalent among the cannabis users who smoked cigarettes [203], providing initial insight into the possible coexisting mechanism on cognitive profile. Independent use of cannabis was also associated with delayed retrieval and low verbal efficiency [204]. However, contradiction persists, excluding THC from impairing verbal memory retrieval [136]. Cannabidiol administration in memory-impaired rats improved memory retrieval in object identification [140]. Oral CBD intake also improved the cerebral blood flow to the hippocampus and regions associated with memory advancement, indicative of improved overall memory function [205], creating further scope for cannabidiol in neurological impairments. The evidence of cannabinoid interaction with respect to retrieval is limited, but it is now evident that cannabinoid agonist and THC negatively impact the process while CBD positively modulates memory retrieval; however, the existing contradictions warrant further studies.

## 6. Current Legal Status: Cannabis India

The increasing decriminalization of cannabis internationally has increased awareness within the Indian Government. Presently, cannabis use in India is regulated in coherence with the Narcotic Drugs and Psychotropic Substances Act (NDPS), 1985, which prohibits the cultivation, possession, manufacturing, and sale of cannabis as Charas (resin), Hashish (liquid form), and Ganja (flowering or fruiting tops) excluding leaves or seed (Bhang) from the purview of the act [206]. However, provisions for cultivation exist but are strictly restricted for research purposes, excluding medical purposes owing to its limited proven use [207]. Recently, in 2013, there was an allotment of regulatory provisions for phytochemicals, followed by the phytochemical act in 2015, creating a new category for plant-based products and encouraging research commitments [208,209,210]. The concept of a medicinal cannabis program was already in progress when in 2016, BOHECO (Bombay Hemp Company) and the Council of Scientific and Industrial Research (CSIR) collaboratively organized ICARE (India Cannabis Analysis Research Education) to build awareness. In the following year, the Government of India approved its first-ever license to Council of Scientific and Industrial Research–Indian Institute of Integrative Medicine (CSIR–IIIM) in collaboration with BOHECO to grow and formulate cannabis-based medicines [211]. India has no approved medicinal cannabis product to date; however, in 2018, the Central Council for Research in Ayurvedic Sciences (CCRAS) reported the pioneer cannabis clinical study where it reduced pain in cancer patients post chemotherapy and radiotherapy [212]. Presently, IIIM Jammu cultivates cannabis and assesses the medicinal components for cancer, epilepsy, and sickle cell anemia, seeking approval for future clinical trials [213]. Moreover, recently, India, during the 63rd United Nations (UN) Commission on Narcotic Drugs (CND) session, voted in favor of removing cannabis and its resin from the UN narcotic drug list [214]. The leniency in the decision from India in favor of cannabis legalization during the UN session indicated a future medicinal cannabis program in India.

Moreover, with the growing interest of scientists worldwide in venturing into the medicinal properties of cannabis, various derivatives of cannabis are in trial, as described in Table 1, as it is already regulated in countries of Latin America, North America, and Europe [215]. Among the list, the United States has been the first to approve the medicinal cannabis later 33 other states also approved use of cannabis for specified medical conditions. [216]. In addition, beneficial effects of cannabidiol were well demonstrated in various preclinical models [217,218,219], which offer hope in the fields of neurological and pain translational research.

Another study, by Thompson et al. (2020), demonstrated AD disease-related changes in the endocannabinoid system. Additionally, the study reflected upon the cross-talk between the endocannabinoid system with M1 muscarinic acetylcholine receptors (mAchR), and thereby highlighted the role of the endocannabinoid system as a promising target in the alleviation of cognitive deficits in AD [220].

The outcomes described above emphasize the importance of bringing reforms in the cannabis drug policy, thus permitting further research for identifying concrete evidence regarding human use. Recently, a systematic review and meta-analysis were performed by reviewing 41 research articles. The results obtained after studying a variety of datasets and using multiple statistical techniques demonstrated that decriminalizing or legalizing cannabis usage did not increase the juvenile use of the drug [221]. There was another study conducted in elderly population to explore the medicinal properties of cannabis, and its utilization in elderly population [222].

**Table 1 biomolecules-13-00162-t001:** Current regulatory status of cannabis-based formulations and their action.

Product	Current Status	Uses	References
Sativex (Nabiximols)	Approved in Brazil, Columbia, Chile, and United Arab Emirates	Neuropathic pain, multiple sclerosis, spasticity	[223,224,225,226]
Epidiolex	Approved by US FDA, 2018	Idiopathic epilepsy, CDKL5 deficiency, resistant Lennox–Gastaut syndrome, Dravet syndrome, tuberous sclerosis complex	[227,228,229,230]
Dronabinol (Marinol, Syndros, REDUVO, and Adversa)	Approved by US FDA	Pain in MS, adjunct therapy in Alzheimer’s, Parkinson–dystonia, acute pain management, HIV complications	[231,232,233,234,235]
ZYN002 (CBD gel)	Under open-label phase 2 assessment, USA	Fragile X syndrome, osteoarthritis	[236,237]
PTL101	Under phase 2 assessment, Israel	Pediatric intractable epilepsy	[238]
CT-921	Preliminary animal testing, human trial awaited, Canada	Neuropathic pain	[239]
Oral CBD solution	Phase 2 trials (20–40 mg) phase 3 trial (adjunct therapy), USA	Resistant seizure disorders; Prader–Willi syndrome	[240,241]
Synthetic crystalline powder of CBD (capsule 200–800 mg)	Phase 2 study, USA	Cannabis-induced personalized effects	[242]
Arvisol	Under phase I trial, Germany	Intended to treat schizophrenia and epilepsy	[241]

## 7. Conclusions

THC is responsible for detrimental effects on the cognitive domain, whereas CBD attenuates the neurological disorders or deficits caused by THC [243,244]. CBD or CBD/THC differently affects learning, consolidation, and retrieval. The combination [244] can come forth as a mainstream drug for neurocognitive disorders; however, continuous research is essential. Owing to various legal constraints concerning cannabis, a setback has occurred in the interest of researchers. A few researchers have revealed the negative effects of CBD, undermining the value of the phytocannabinoids (CBD). Such differences can only be eradicated with extensive research. The present review highlights the therapeutic use of cannabis following the establishment of proper dose, site, and frequency of drug administration.

## Figures and Tables

**Figure 1 biomolecules-13-00162-f001:**
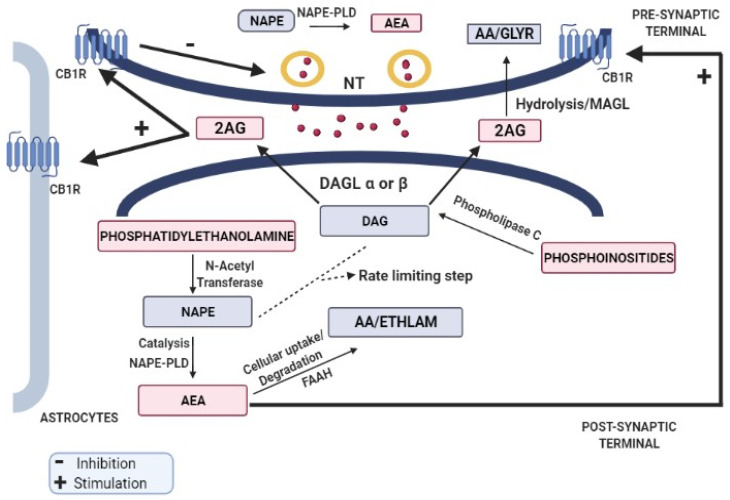
Endocannabinoid retrograde signaling. The endocannabinoids act through a retrograde signaling pathway where cannabinoids are released from depolarized postsynaptic neurons via calcium-dependent mechanism and act in a retrograde fashion on the presynaptic CBRs to inhibit neurotransmitter release.

## Data Availability

Data that support the findings of this study are available in standard research databases such as PubMed, Science Direct, or Google Scholar, and/or on public domains that can be searched with either keywords or DOI numbers.

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
