# Peer review of "Effect of Cannabis on Memory Consolidation, Learning and Retrieval and Its Current Legal Status in India: A Review"

_biomolecules, 2023, doi:10.3390/biom13010162_

Round 1
Reviewer 1 Report
This is an interesting review article that discusses recent literature on the pharmacological effects of cannabis and its ramifications for learning and memory as well as its legal status. The review is timely with the rising worldwide acceptance of cannabis. The authors discuss interesting historical and religious information about cannabis and its use in ancient civilizations. The authors also discuss the endogenous cannabinoid system, cannabinoid receptors, actions of cannabis constituents on cognitive functions such as learning, memory consolidation and memory retrieval. Finally, the review discusses the current legal status of cannabis in India and the regulatory status of cannabis-based formulations. The review is interesting and comprehensive. Recent data on the following points are suggested to be discussed to improve the quality of the manuscript.
1- Recent data on central and peripheral cannabinoid receptors localization.
2- Cannabidiol interaction with NavMs voltage-gated sodium channels as well as activation of neuronal M-currents via activation of neuronal Kv7 channels.
3- Recent literature on the behavioral phenotypes of CB1 knockout mice such as learning and memory deficits.
4- Which CB receptors in the cerebellum are essential for cerebellar learning?
5- Avet et. al. have recently shown cross-talk between cannabinoid and muscarinic receptors, transactivation of CB1 and CB2 receptors by acetylcholine and endocannabinoid release in the hippocampus by activation of muscarinic acetylcholine receptors. Please discuss this point and its potential impact on legal status of cannabis.
6- Age-related effects as preclinical data suggest that Δ9-tetrahydrocannabinol effects are more pronounced during adolescence.
7- Please discuss the recent literature and clinical data indicating that Δ9-tetrahydrocannabinol influences false-memory.
8- The roles of central and peripheral cannabinoid receptors -in adrenergic and noradrenergic cells- in stress-induced memory impairment.
9- Modulation of memory consolidation by endogenous cannabinoids in the prefrontal-limbic circuit.
English language grammatical errors are present throughout the manuscript. English should be reviewed by a native English speaker. I list below only few examples:
The word “endocannabinioids” is spelled incorrectly throughout the manuscript.
Line 80 “as an herb of “please review English.
Line 90 “in such a fashion repeating the effects produced by” this sentence is not clear.
Line 91 “Apart from the presence other endogenous substances” this sentence is not clear.
Line 94 “whereas produces inactivity at CB2 receptor” please review English.
Line 105 “Figure1” review space between words here and elsewhere in the manuscript.
Line “112” “the inhibition neurotransmitter release” review English.
Other comments
Figure 1 “Endocannabinoid retrograde signaling” please mention software used to prepare the figure.
Reviewer 2 Report
This review focuses on the impacts of cannabis and its constituents on learning and memory, and addresses the current state of legality of the herb in India. This topic is likely of greatest interest to those who are practitioners or are involved in making policy decisions in that country. The review is thorough, and would be appropriate for publication after the following issues are addressed.
* extensive review of English grammar and spelling is needed.
* some passages are particularly difficult to understand, as written: lines 90-91 say "AEA...produces inactivity at [the] CB2 receptor", however AEA is most accurately classified as a CB2 partial agonist (see PMID 10779390). Line 235-236 (p.6) says "extreme cannabis exposure"; what does "extreme" mean? Please be more specific/scientific. The last sentence on p. 6 (lines 274-276) is difficult to understand.
* p. 5, line 219: should CA1 be "CB1"?
* p. 9, paragraph starting line 404: it should be mentioned explicitly that CBD (as Epidiolex(R)) is approved by the United States FDA; this is posted in the table, but should also be mentioned in the text where preclinical assessment of CBD is discussed. Table 1: the current status column should specify which country has approved each product, or where preclinical trials are being performed.
* conclusions p. 10, lines 425-426: what are "conflicting findings" that are "undermining the value of CBD"? CBD has been extensively studied in preclinical tests and has been found to be safe and effective. This should be softened.
Round 2
Reviewer 1 Report
The authors have addressed the comments.